# Influences of Wood Decomposition Associated with Tree Types on Soil Nutrient Concentrations and Enzyme Activities

Xiang-Yu Ji [1,2], Qian Xu [1,2], Zhu-Qi Zhao [1,2], Yu-Xiong Zheng [1,2], Lei Deng [2,3] and Zhen-Hong Hu [2,3,*]

1   College of Natural Resources and Environment, Northwest A&F University, Yangling 712100, China;
    xy_2022@nwafu.edu.cn (X.-Y.J.); qianxu@nwafu.edu.cn (Q.X.); zzq_juicy@nwafu.edu.cn (Z.-Q.Z.);
    yxzheng@nwafu.edu.cn (Y.-X.Z.)

2   State Key Laboratory of Soil Erosion and Dryland Farming on the Loess Plateau, Northwest A&F University,
    Yangling 712100, China; leideng@ms.iswc.ac.cn

3   Institute of Soil and Water Conservation, Chinese Academy of Sciences and Ministry of Water Resources,
    Yangling 712100, China

*   Correspondence: zhhu2020@nwafu.edu.cn

**Abstract:** Wood decomposition is a biogeochemical process fundamental to element cycling in forest ecosystems, which could alter the nutrient concentrations and enzyme activities of the underlying forest soils. Wood traits, which vary by tree species, can influence decomposition aboveground, but it is not well understood how wood decomposition associated with different tree types (i.e., angiosperm and gymnosperm species) influences underlying soil nutrient concentrations and enzyme activities. In this study, we evaluated how tree type (for four angiosperm vs. four gymnosperm species) affects underlying soil total carbon (C), nitrogen (N), and phosphorus (P) concentrations; microbial biomass C, N, and P concentrations; and C-, N-, and P-acquiring enzymes activities. We found that decomposing wood significantly increased soil total P, and microbial biomass C and P concentrations. However, the differences in the nutrient concentrations of soil and microbial biomass beneath decomposing wood were not different between angiosperm and gymnosperm species. Surprisingly, the activities of soil C-, N-, and P-acquiring enzymes beneath the decomposing wood differed significantly between angiosperm and gymnosperm species. The soils beneath decomposing angiosperm wood had higher P-acquiring enzyme activity, while the soils beneath gymnosperm wood had higher C- and N-acquiring enzyme activities. The soils beneath angiosperm and gymnosperm wood had a similar C-limitation for microbial metabolism, but the microbial metabolism in soils beneath angiosperm wood was more P-limited compared to soils beneath gymnosperm wood. In conclusion, our findings highlight that the tree types of decomposing wood may affect underlying soil enzyme activities and enzyme characteristics, improving our ability to accurately predict the role of wood decomposition on forest nutrient cycles.

**Keywords:** wood decomposition; tree species; tree types; forest soils; soil nutrient; soil enzyme





## 1. Introduction

Deadwood represents a substantial carbon (C) pool, as it accounts for ~10%–20% of the total biomass C in global forest ecosystems [1]. Wood decomposition could release dissolved organic matter and return nutrients to the underlying soils [2,3] and further affect soil microbial characteristics [4,5]. Thus, wood decomposition is a biogeochemical process fundamental to element cycling in forest ecosystems. For example, under three decomposing eucalypt (including *Eucalyptus melliodora* A. Cunningham ex Schauer, *E. blakelyi* Maiden, and *E. macrorhyncha* F. Mueller ex Bentham) woods, soil total C, nitrate-nitrogen, and available phosphorus (P) concentrations increased, while soil pH was reduced [6]. Moreover, a study has shown that concentrations of soil organic C, extractable C, and microbial biomass C and nitrogen (N) concentrations increased under *Picea abies* (Linnaeus) H. Karsten wood, along with glucosidase activity [7]. It was also reported that C and

N released by decomposing *P. abies* wood were positively correlated with soil C- and N-acquiring enzyme activities [8].

Tree species vary greatly in wood traits such as nutrient contents [9]. Wood from angiosperm species generally contains higher contents of N and P than wood from gymnosperm species [10]. These differences are reflected in higher decomposer activities and decomposition rates for angiosperm wood [11–13]. Thus, the tree types (i.e., angiosperm and gymnosperm species) of decomposing wood may strongly influence the nutrient cycles of underlying soils. This study aims to discover and explain the influences of wood decomposition associated with tree types on soil nutrients and enzymes, as it is not well understood. A few studies found that coniferous (i.e., gymnosperm species) forests have a more acidifying effect on the soil compared to broadleaved (i.e., angiosperm species) forests [14–17]. The differences in leaf litter traits (e.g., its aluminum (Al) content and degradability) could explain the higher soil acidity of coniferous forests [16]. The higher lignin content of gymnosperm litter also hampers its decomposition and leads to the accumulation of organic carbon, which may acidify soils [18]. It was also reported that the ratio of unstable C to total organic C of secondary coniferous forest soils was lower and the microbial quotient was higher compared to that of broadleaf forests, indicating that the plant litter of gymnosperm species could be less utilized by soil microorganisms compared to that of angiosperm species [19]. Moreover, several studies have shown differences in soil nutrient concentrations under the decomposing wood of different tree types. For example, angiosperm wood released much more dissolved organic C and N than gymnosperm wood during decomposition [20]. It was also reported that wood C, N, and P contents varied with tree types (including three angiosperm and one gymnosperm species), and that wood nutrient contents were correlated with underlying soil nutrient concentrations [21]. However, a study has also shown that the wood decomposition of different tree types (including nine angiosperm and four gymnosperm species) did not lead to significant differences in soil nutrient concentrations or microbial biomass [22].

Soil extracellular enzymes catalyze the mineralization of soil organic matter, releasing nutrients, such as C, N, and P, back into the environment [23]. Abiotic factors, such as soil pH, soil temperature, soil moisture, and climate conditions, were found to regulate soil enzyme activities [23–27]. It was confirmed that the main regulatory mechanism of these factors on soil enzyme activities is the altered kinetic parameters of enzyme-catalyzed reactions [28,29]. Moreover, resource allocation theory posits that as nutrient availability in the environment increases, soil microorganisms reduce their investment in corresponding enzymes, leading to a decrease in enzyme activities, and vice versa [30–32]. For example, N addition was found to reduce N-acquiring enzyme activity in temperate soils, which are considered N-limited [33–35], and P addition was found to greatly reduce the activity of P-acquiring enzyme activity in tropical and subtropical soils, which are considered P-limited [36–40]. A meta-analysis at the global scale also showed that P addition significantly reduces soil P-acquiring enzyme activity [41].

Soil enzyme activities may reflect information on environmental resource limitations for microbial metabolism, as the microbial investment in soil extracellular enzymes is regulated by environmental resource availability [30–32]. A global-scale meta-analysis revealed that the logarithmic ratios of soil C-, N-, and P-acquiring enzyme activity are nearly 1:1:1 across ecosystems, likely indicating a natural boundary of C, N, and P availability around soil microorganisms [23]. The method of the logarithmic ratio of enzyme activities was retained and further developed into the enzyme vector model, which focuses on the locus reflecting information on the logarithmic ratio of both the x- and y-coordinates of a Cartesian coordinate and calculates its distance (i.e., the vector length) and angle (i.e., the vector angle) from the origin [42]. Recently, the statistically optimized enzyme vector model has become widely used, quantifying relative C, N, and P limitations for microbial metabolism through vector angles and vector lengths, which are calculated on the basis of enzyme activity [43,44]. The model calculates the x-coordinate of the locus using C/(C + P)-acquiring enzyme activities and calculates the y-coordinate of the locus

using C/(C + N)-acquiring enzyme activities. Thus, if the logarithmic ratios of soil C-, N-, and P-acquiring enzyme activity is 1:1:1 as natural, the locus has a length of 0.71 and an angle of 45° [23,45]. As P-acquiring enzyme activity increases, indicating that microbial metabolism is P-limited, the angle degree of the locus increases. Conversely, as N-acquiring enzyme activity increases, indicating that microbial metabolism is N-limited, the angle degree of the locus decreases. Thus, a vector angle degree > 45° represents P-limitation, while a degree < 45° represents N-limitation.

Tropical soils are more prone to P deficiency due to strong weathering, leaching, and loss processes, as well as the accumulation of occluded P [46]. This limits the growth of plants [47] and soil microorganisms [48], and may put greater pressure on organic P mineralization [36]. For example, in forest ecosystems across four climatic zones (21°47′–42°24′ N) in eastern China, as the degree of P limitation increased, the catalytic efficiency of litter and soil P-acquiring enzymes rapidly increased [49]. The subtropical forest had the highest degree of P limitation, and thus the maximum reaction rate and catalytic efficiency of litter and soil P-acquiring enzymes were also the highest in the subtropical forest [49].

This study assessed the effects of decomposing angiosperm and gymnosperm wood on the underlying soil nutrient concentrations and enzyme activities in a subtropical forest. We evaluated two hypotheses: (1) due to the higher N and P contents of angiosperm wood compared to gymnosperm wood [10], and the higher amount of released dissolved organic C and N during the decomposition of angiosperm wood compared to gymnosperm wood [20], the tree types of decomposing wood could affect underlying soil nutrient concentrations, such as total C, N, and P and microbial biomass C, N, and P concentrations; and (2) due to the differences in soil nutrient concentrations caused by wood decomposition associated with tree types (i.e., hypothesis 1), and the regulatory effect of soil nutrient concentrations on enzyme activities [30–32], the tree types of decomposing wood could affect underlying soil enzyme activities, such as C-, N-, and P-acquiring enzyme activities.

## 2. Materials and Methods

### 2.1. Study Site

This study was carried out at the Tiantong National Forest Ecosystem Observation and Research Station (29°48′ N, 121°47′ E, and an elevation of 160 m), in Zhejiang Province, China. The study site exhibits a characteristic subtropical monsoon climate characterized by hot and humid summers, as well as dry and cold winters [50]. The study site experiences an average annual temperature of 16.2 °C, with monthly mean air temperatures ranging from 4.2 °C in January to 28.1 °C in July. The region receives an average annual precipitation of 1374 mm, with the majority of rainfall occurring between May and August [51]. The main soil type in the area was identified as Ferric Acrisol [52], with a soil pH ~4.9 [53]. The forest is dominated by typical subtropical evergreen broad-leaved vegetation, including *Castanopsis fargesii* Franchet, *Schima superba* Gardner & Champion, *Cunninghamia lanceolata* (Lambert) Hooker, *Machilus thunbergii* Siebold & Zuccarini, and *Neolitsea aurata* (Hayata) Koidzumi.

### 2.2. Experimental Design

In October 2017, wood samples were obtained by harvesting the trees of eight species, including four angiosperm species (*S. superba*, *Michelia maudiae* Dunn, *C. fargesi,* and *Liquidambar formosana* Hance) and four gymnosperm species (*C. lanceolata*, *Pseudolarix amabilis* (J. Nelson) Rehder, *Cryptomeria fortunei* Hooibrenk ex Billain, and *Pinus massoniana* Lambert), which are dominant species in the local forest [11]. All wood samples used in this study were obtained from plantation forest trees of similar ages growing in the same stand, except for *C. fargesi*, which were obtained from non-plantation forest trees, as part of the forest at the study site was replanted following logging activities in the 1960s [11]. Three stems of each species were felled, and one 1.1 m section (including bark) was obtained from each stem. Two 5 cm thick disks were cut from both ends of each stem section and transported to the laboratory for analysis. We analyzed the chemistry of the mixture of bark, sapwood, and heartwood from each stem disk (Table 1). Bark samples

obtained from each disk were ground into a fine powder using a ball mill. Sapwood and heartwood samples were obtained using an electric drill with an 8 mm drill bit that was sterilized with ethanol between samples. Each wood sample comprised at least 15 drill holes. Then, we measured wood C and N contents by high-temperature oxidation using an elemental analyzer (Vario MACRO cube, Elementar, Germany). We measured wood P content using the molybdenum antimony colorimetric technique [54], which digests wood mixtures using $H_2SO_4$ and $HClO_4$, and the extract was analyzed using an ultraviolet-visible spectrophotometer (UV2450, Shimadzu, Kyoto, Japan).

**Table 1.** The nutrient contents of angiosperm and gymnosperm wood. Mean values and standard errors are shown. Different letters indicate significant differences between angiosperm and gymnosperm wood ($p < 0.05$).

| Nutrient Content (g kg$^{-1}$) | Tree Type | | p Value |
| --- | --- | --- | --- |
| | Angiosperm | Gymnosperm | |
| Wood C | 327 (28) b | 403 (17) a | 0.033 |
| Wood N | 3.6 (0.2) a | 2.3 (0.2) b | <0.001 |
| Wood P | 0.35 (0.05) a | 0.16 (0.01) b | 0.002 |

This study used three 20 m × 20 m plots. Wood samples (including bark) of a length of 1 m were arranged in a randomized order within each plot and spaced over 1 m apart. The mean diameters of these stem sections were recorded as 13.52 ± 0.29 cm and 13.80 ± 0.17 cm for angiosperm and gymnosperm species, respectively. PVC panels measuring 3 mm in thickness were driven 60 cm into the ground to prevent topsoil nutrients from flowing sideways with water. The period of wood decomposition lasted 3 years.

*2.3. Soil Sampling and Measurements*

In October 2020, we collected soil cores of 0–10 cm depth away from decomposing wood (the bare soil) and soils beneath the angiosperm and gymnosperm wood using a stainless-steel sampler with a diameter of 5 cm. To ensure representative samples, three soil replicates were collected beneath each wood stem section, and these replicates were combined into one composite sample. In addition, three soil cores were randomly collected from each of the 3 plots, at least 4 m away from the edge of the wood stem section, and these replicates were combined into composite samples. We collected a total of 27 soil samples ((1 bare soil + 8 soils beneath the decomposing wood) × 3 replicates) and kept the samples refrigerated on ice packs during transport to the laboratory. After sieving soils through a 2 mm mesh, we divided each sample into two parts: one was refrigerated at −20 °C for the analyses of soil enzyme activities and soil microbial biomass C, N, and P concentrations; and the other was air-dried for the analyses of soil total C, N, and P concentrations.

We measured soil microbial biomass C, N, and P concentrations using the chloroform fumigation and potassium sulfate extraction techniques [55–57]. The calculation of soil microbial biomass C, N, and P concentrations utilized conversion factors of 0.45, 0.45, and 0.4, respectively [55–57]. We measured soil total C and N concentrations from the air-dried soil samples by high-temperature oxidation using an elemental analyzer (Vario MACRO cube, Elementar, Germany). Soil samples were further ground and sieved through a 0.15 mm mesh to determine total P concentration. We measured soil total P concentration using the molybdenum antimony colorimetric technique [54], which digests soil samples using $H_2SO_4$ and $HClO_4$, and the extract was analyzed using an ultraviolet-visible spectrophotometer (UV2450, Shimadzu, Japan).

We followed German et al. [58] to measure the potential activity of four main hydrolytic soil enzymes, including two C-acquiring enzymes (β-1,4-glucosidase (BG; EC 3.2.1.21) and cellobiohydrolase (CB; EC 3.2.1.91); one N-acquiring enzyme (β-1,4-*N*-acetyl-glucosaminidase (NAG; EC 3.2.1.14)); and one P-acquiring enzyme (acid phosphatase (AcP; EC 3.1.3.1)). These enzymes play fundamental roles in the biogeochemical process of C,

N, and P: BG hydrolyzes cellobiose to glucose; CB hydrolyzes cellobiose dimers; NAG hydrolyzes chitin and other glucosamine polymers; and AcP hydrolyzes phosphomonoesters to phosphate. Moreover, the activities of these enzymes could be further calculated using the vector length and vector angle based on the enzyme vector model.

Soil homogenates were prepared by shaking 1 g of the fresh soil sample with 125 mL of water at 180 r/min and 25 °C for 2 h. The samples were then mixed with 250 μL of a 200 μM fluorescent substrate solution, water, and 1 μM 4-methylumbelliferone (MUB) solution, respectively, to prepare the assay sample solution, homogenate control solution, and quench control solution. The fluorescent substrates were 4-MUB-β-D-glucoside, 4-MUB-β-D-cellobioside, 4-MUB-*N*-acetyl-β-D-glucosaminide, and 4-MUB-phosphate, used for BG, CB, NAG, and AcP, respectively. After incubation in the dark at 25 °C for 4 h, 50 μL of 0.5 M NaOH solution was added to stop the reaction. The substrate control solution and standard sample solution were configured according to a ratio of 6 mL water, 180 μL 0.5 M NaOH solution, and 1.5 mL 200 μM fluorescent substrate solution or 1 μM MUB solution. A 250 μL solution of each sample was added into the well of black 96-well microplates and fluorescence was measured using a microplate reader (VictorX, PE, Waltham, MA, USA) with 365 nm excitation and a 450 nm emission filter. The unit of enzyme activity was nmol $g^{-1}$ $h^{-1}$.

According to the enzyme vector model [42,44], we calculated the vector angle and vector length of the soil enzymes:

$$\text{Vector Angle}(°) = \tan^{-1}\frac{y}{x} \times \frac{180}{\pi}$$

$$\text{Vector Length} = \sqrt{x^2 + y^2}$$

where values of x represent (BG + CB)/(BG + CB + AcP) and values of y represent (BG + CB)/(BG + CB + NAG). Higher vector angle degrees represent more likely limitations to microbial metabolism by soil P availability, and larger vector lengths represent more likely limitations to microbial metabolism by soil C availability. In addition, the enzyme vector model assumes that the boundary between N and P limitations is the vector angle of 45° [42].

*2.4. Data Analysis*

All statistical analyses were performed using the statistical software R 4.2.2 [59]. Data are presented as mean values ± standard error (n = 3 for bare soil and n = 12 for soils beneath angiosperm and gymnosperm wood). Before performing further data analysis, the Shapiro–Wilk test was used to determine whether the data conformed to a normal distribution; otherwise, the data were log-transformed to meet the assumption of homoscedasticity, which is necessary for the following statistical methods. One-way ANOVA was used to determine the effect of decomposing wood on soil total C, N, and P concentrations, microbial biomass C, N, and P concentrations, and soil C-, N-, and P-acquiring enzyme activities. Bonferroni's least significant difference (LSD) test was used to compare the mean values between the bare soil, the soil beneath angiosperm wood, and the soil beneath gymnosperm wood, using the "agricolae" package (v1.3.5) [60]. Moreover, Bonferroni's LSD test is considered to be slightly more conservative than other LSD tests (e.g., Tukey's honest significance test), but it was more appropriate for this study as the sample sizes of the three soil types were different. T-tests were used to compare the mean values for soils beneath the angiosperm and gymnosperm wood. A principal component analysis (PCA) was conducted to visualize the overall variation in soil enzymes [23] using the "vegan" package (v2.6.4) [61]. Soil enzyme activities are correlated with each other sometimes, and thus PCA is suitable for reducing their dimensionality. Mean species scores of the first two PCA axes were calculated to represent enzyme activity patterns, as the proportion of variance explained by the first two PCA axes was relatively high. All figures were generated using the "ggplot2" package (v3.4.1) [62].

## 3. Results

### 3.1. Soil Total C and Nutrient Concentrations

Wood decomposition significantly increased the total P concentration of underlying soils for both angiosperm and gymnosperm wood (both $p < 0.05$; Figure 1c) and the C concentration for gymnosperm wood ($p < 0.05$; Figure 1a). However, the differences in soil total C and P concentrations were not significantly different between angiosperm and gymnosperm wood (both $p > 0.05$; Figure 1a,c). Additionally, wood decomposition significantly increased microbial biomass C (both $p < 0.05$; Figure 1d) and microbial biomass P concentrations (both $p < 0.05$; Figure 1f) of the underlying soils, but the differences were not significant between the angiosperm and gymnosperm wood (both $p > 0.05$; Figure 1d,f). Finally, there were no significant differences in the soils with respect to soil total N, microbial biomass N concentrations (both $p > 0.05$; Figure 1b,e), or the soil C/N ratio ($p > 0.05$; Figure S1).

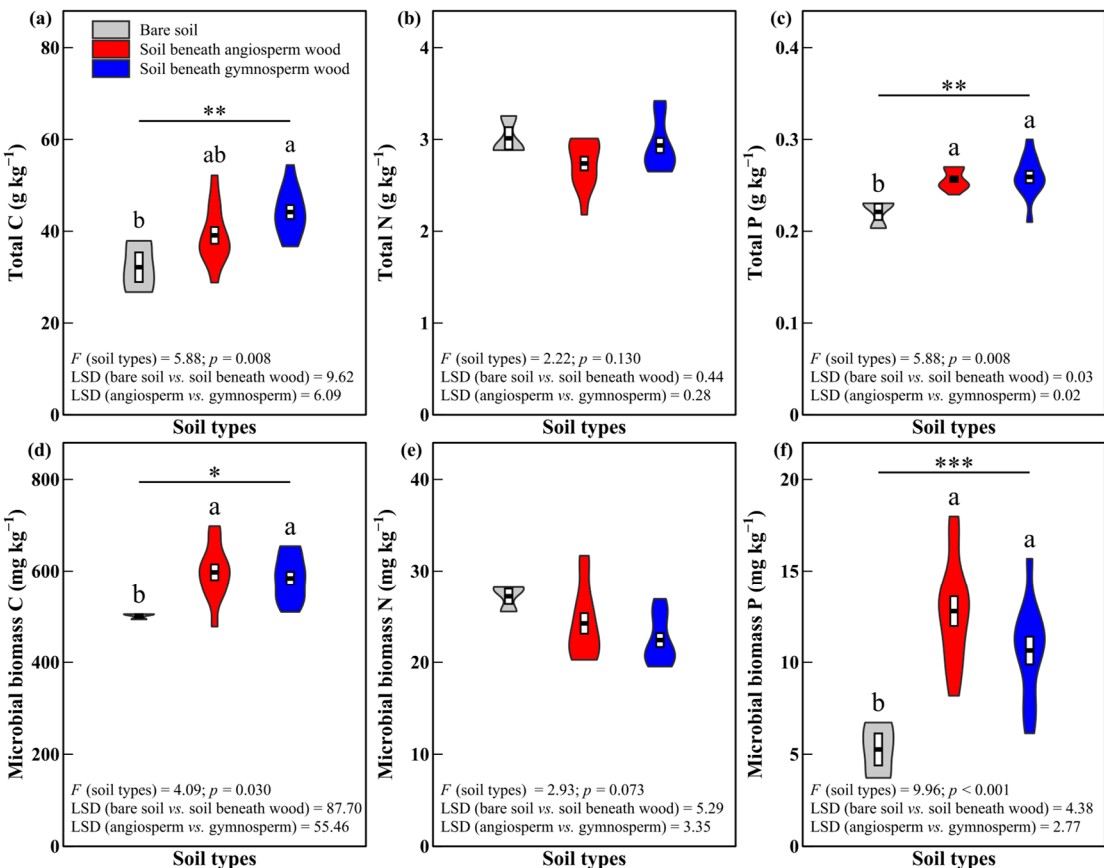

**Figure 1.** Effects of decomposing wood on soil total C and nutrient concentrations. The crossbar within violin plots shows mean values ± standard error ($n_{Bare\ soil} = 3$, $n_{Soil\ beneath\ angiosperm\ wood} = n_{Soil\ beneath\ gymnosperm\ wood} = 12$) of total C (**a**); total N (**b**); total P (**c**); microbial biomass C (**d**); microbial biomass N (**e**); and microbial biomass P (**f**) concentrations of bare soil and soils beneath angiosperm and gymnosperm wood. *, $p < 0.05$; **, $p < 0.01$; ***, $p < 0.001$. Different letters indicate significant differences among soil types ($p < 0.05$).

### 3.2. Soil Enzyme Activities and Enzyme Characteristics

Wood decomposition significantly influenced the enzyme activities of the underlying soils (Figure 2). In contrast with the limited effect on soil nutrients, the tree types of decomposing wood strongly affected soil enzyme activities. The activity of β-1,4-glucosidase (BG), N-acetyl-glucosaminidase (NAG), and acid phosphatase (AcP) differed significantly for the soils beneath decomposing angiosperm and gymnosperm wood (Figure 2a,c,d). In terms of BG and NAG activities, the soil beneath gymnosperm wood showed significantly

higher activities than those beneath angiosperm wood (both $p < 0.05$; Figure 2a,c). The activity of AcP in soil beneath angiosperm wood was significantly higher than that beneath gymnosperm wood ($p < 0.05$; Figure 2d). In contrast to the other three enzymes, cellobiohydrolase (CB) activity did not differ significantly between the soils beneath decomposing angiosperm and gymnosperm wood ($p > 0.05$; Figure 2b).

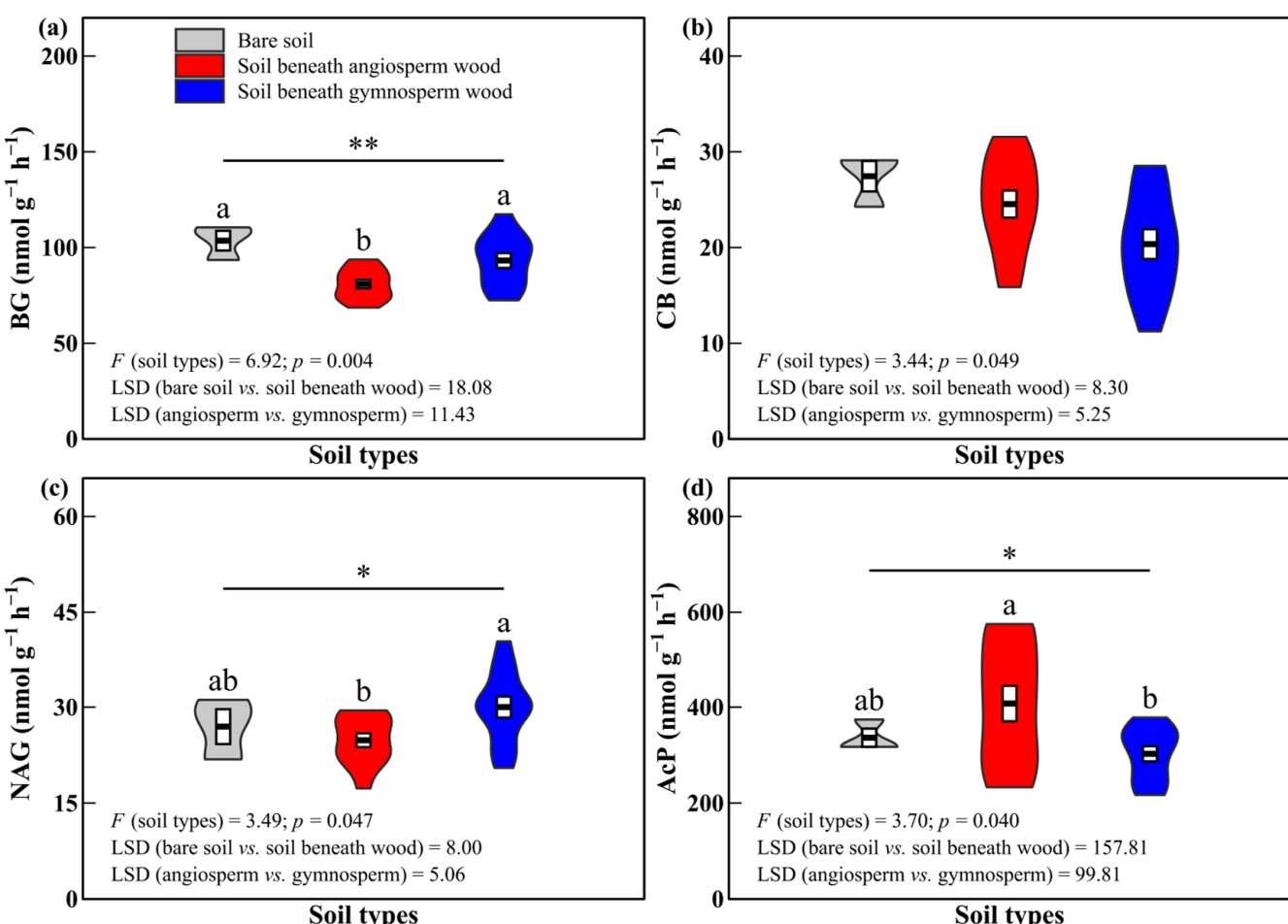

**Figure 2.** Effects of decomposing wood on soil enzyme activities. The crossbar within violin plots shows mean values $\pm$ standard error ($n_{Bare\ soil}$ = 3, $n_{Soil\ beneath\ angiosperm\ wood}$ = $n_{Soil\ beneath\ angiosperm\ wood}$ = 12) of BG (**a**); CB (**b**); NAG (**c**); and AcP (**d**) activities of bare soil and soils beneath angiosperm and gymnosperm wood. BG: β-1,4-glucosidase; CB: cellobiohydrolase; NAG: β-1,4-N-acetyl-glucosaminidase; AcP: acid phosphatase. *, $p < 0.05$; **, $p < 0.01$. Different letters indicate significant differences among soil types ($p < 0.05$).

The variation in soil enzyme activities was relatively well captured by PCA analysis, with the first two PCA axes together representing 77.32% of the variation (Figure 3a). The first PCA axis accounted for 44.20% of the overall variation: CB and AcP contributed substantially to the first PCA axis, as observed by the high individual loadings (loadings > 0.60; Table S1). The second PCA axis explained 33.12% of the total enzyme variation: BG contributed substantially to the second PCA axis, as observed by the high individual loading (Table S1). Moreover, NAG also contributed to both the first and the second PCA axes (for both axes, loadings > 0.50; Table S1).

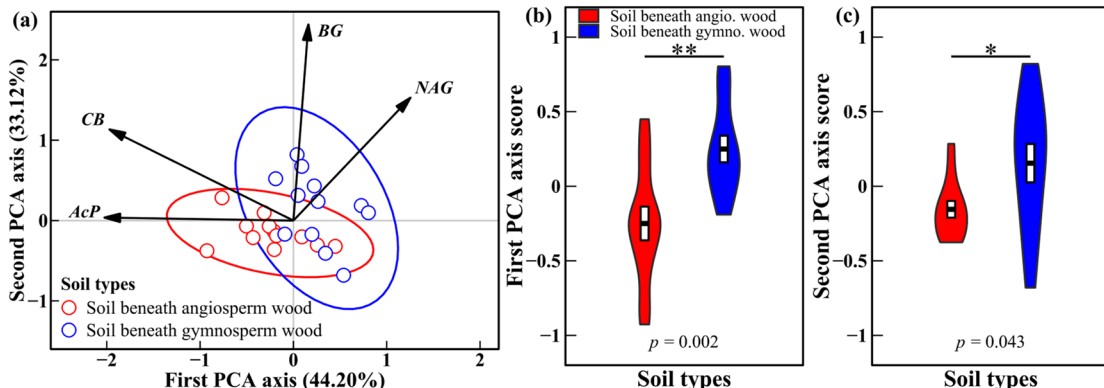

**Figure 3.** Principal component analysis (PCA) of soil enzymes beneath decomposing angiosperm and gymnosperm wood (**a**); ellipses show the 95% confidence interval for the respective tree types. The crossbar within violin plots shows mean values ± standard error ($n_{Soil\ beneath\ angiosperm\ wood}$ = $n_{Soil\ beneath\ angiosperm\ wood}$ = 12) of the first PCA axis (**b**); and second PCA axis (**c**) scores of soils beneath decomposing angiosperm and gymnosperm wood. BG: β-1,4-glucosidase; CB: cellobiohydrolase; NAG: β-1,4-N-acetyl-glucosaminidase; AcP: acid phosphatase. *, $p < 0.05$; **, $p < 0.01$.

The soil enzymes beneath angiosperm and gymnosperm wood clustered close to the center, but the mean scores of the first two PCA axes differed significantly between the soils beneath decomposing angiosperm and gymnosperm wood (for the first PCA axis, $p < 0.01$; for the second PCA axis, $p < 0.05$; Figure 3b,c). This indicates that the tree types of decomposing wood strongly influenced the enzyme activity patterns of the underlying soil. Moreover, the vector angle of soil enzymes beneath angiosperm wood was significantly higher than that beneath gymnosperm wood ($p < 0.01$; Figure 4a), but the difference in vector length was not significant ($p > 0.05$; Figure 4b).

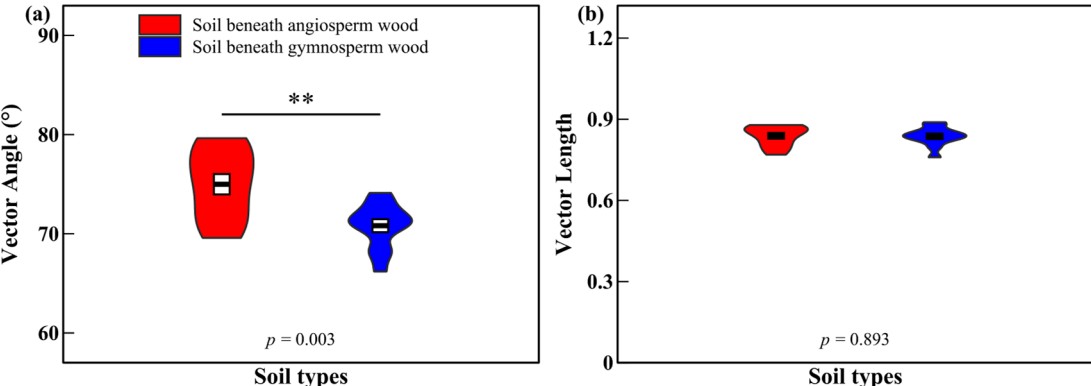

**Figure 4.** Effects of decomposing wood on vector angle and vector length of soil enzymes based on the enzyme vector model. The crossbar within violin plots shows mean values ± standard error ($n_{Soil\ beneath\ angiosperm\ wood}$ = $n_{Soil\ beneath\ angiosperm\ wood}$ = 12) of soil enzyme vector angle (**a**); and vector length (**b**) beneath decomposing angiosperm and gymnosperm wood. **, $p < 0.01$.

## 4. Discussion

### 4.1. Effects of Decomposing Wood on Soil Total C and Nutrient Concentrations

In contrast to our first hypothesis, we observed no significant differences in soil total C or in the nutrient concentrations between soils beneath decomposing angiosperm and gymnosperm wood (Figure 1). Three factors could explain this lack of differentiation. First, both angiosperm and gymnosperm wood may have still been in the initial stages of decomposition during this three-year decomposition experiment, and relatively few nutrients were released from the decomposing wood to the underlying soils. Second, the rapid immobilization of wood nutrients by fungi or other microorganisms in the

initial decay stages may impede the release of nutrients into the soil [22]. Third, the eight species chosen do not show the differences between angiosperms and gymnosperms, and represent only a limited sample of the two taxonomic groups. This suggests that the extent of the study should be expanded to better qualify differences between angiosperm and gymnosperm species. Our findings suggest that tree types for these eight species of decomposing wood had no effect on underlying soil nutrients. Previous studies found that chemical wood traits, which varied among tree species [5,10,11,40,63], could strongly influence wood decomposition rates [64,65] and $CO_2$ fluxes [11,40], but likely have a limited influence on underlying soil properties as well [22]. A previous study based on 30 forest plots and 13 tree species showed that forest plots lead to greater variation in the soil properties beneath decomposing wood compared to tree species, indicating that the tree species of decomposing wood had a limited effect on underlying soil properties [22]. Moreover, although it was reported that both tree species and stages of decomposition affected soil nutrient concentrations and enzyme activities beneath decomposing wood, the stage of decomposition in some studies was much more important than the tree species in affecting soil properties [21,66]. This is because the release of wood decomposition products (e.g., dissolved organic C) is considered to be the main path regulating soil properties, and more dissolved organic C is released beneath both angiosperm and gymnosperm wood at later stages of decomposition. In summary, tree species and tree types may have a limited effect on soil nutrient concentrations beneath decomposing wood, and decomposing wood at later, but not in the initial, decay stages should be regarded as a nutrient source for underlying soils.

### 4.2. Effects of Decomposing Wood on Soil Enzyme Activities and Enzyme Characteristics

Partly consistent with our second hypothesis, our findings show that the differences in soil enzyme activities were different between the soils beneath the decomposing wood of angiosperm and gymnosperm species. Soils beneath angiosperms had higher P-acquiring enzyme activity, while soils beneath gymnosperms had higher C- and N-acquiring enzyme activities. Soil C-, N-, and P-acquiring enzyme activities are considered to be closely related to soil nutrient concentrations, as the soil microbial community allocates more resources to enzymes that target limiting nutrients [30–32]. Many studies have verified that N or P addition treatment can reduce soil N- and P-acquiring enzyme activities, respectively [33–39,41]. However, this mechanism does not fully explain the differences in soil enzyme activities in our findings, as soil total C and nutrient concentrations beneath the decomposing wood did not differ between angiosperm and gymnosperm species (Figure 1). This indicates that the differences in soil enzymes beneath the angiosperm and gymnosperm wood were not regulated by soil nutrients and rejects part of our second hypothesis. Two factors could explain the differences in soil enzyme activities. First, changes in C- and N-acquiring enzyme activities were regulated by substrate type but not by the degree of resource limitation [67–69]. This suggests that there were more available C and N resources in soils beneath angiosperm wood, while the rich unavailable C and N resources (i.e., cellulose, chitin, and peptidoglycan) induced higher C- and N-acquiring enzyme activities beneath gymnosperm wood. Second, we speculate that differences in soil acidity play a crucial role in regulating the enzyme activities beneath the decomposing wood of angiosperm and gymnosperm species, as pH regulates enzyme conformations and activities and the soil pH beneath gymnosperm wood was significantly lower than that beneath angiosperm wood (Figure S2).

A previous study showed that the composition of soil organic matter under decomposing wood varied with tree types (including one angiosperm and one gymnosperm species), and the composition of soil organic matter was correlated with soil pH [3]. Angiosperm and gymnosperm wood release different phenolic matter (e.g., lignin and its decay byproducts) during decomposition, as angiosperm and gymnosperm wood have different types of lignin, and angiosperm wood is decomposed by white-rot fungi while gymnosperm wood is decomposed by brown-rot fungi. It was shown that decomposing

*Fagus sylvatica* Linnaeus (an angiosperm species) wood could remediate the Al-driven acidity of underlying soils, as its releasing phenolic matter transports calcium (Ca) and adsorbs Al [3]. However, decomposing *Abies alba* Miller (a gymnosperm species) wood could further acidify the underlying soil, as its releasing phenolic matter is less reactive compared to that of angiosperm wood and its releasing water-extractable organic matter is acidic [3]. Moreover, a few studies have shown divergent findings on the relationship between soil pH and enzyme activities to date. Soil BG and NAG activities are usually negatively correlated with soil pH [23,70,71], as both enzymes have optimal pH < 7 [23,72]. However, it was also reported that soil BG and NAG activities were positively correlated with soil pH [26]. The difference was attributed to the different soil pH ranges in these studies [26]. Soil AcP activity is negatively correlated with soil pH [23,70,71]. Lower soil pH in natural ecosystems means a higher degree of weathering and P limitation, improving the soil microbial community investment in AcP production [23,71].

We found that soil enzyme activities beneath the angiosperm and gymnosperm wood clustered close to the center, indicating that the tree types of decomposing wood likely had a limited effect on enzyme activity patterns (Figure 3a), aligning with one previous study [22]. However, the mean scores of the first two PCA axes in the soil beneath decomposing wood differed significantly between angiosperm and gymnosperm species (Figure 3b,c). Thus, the underlying soil enzyme activity patterns were affected by the tree types of decomposing wood. Moreover, the soil enzyme vector angle degrees beneath the decomposing wood of angiosperms and gymnosperms were both higher than 45°, and soil beneath angiosperm wood had a higher vector angle degree than that beneath gymnosperm wood (Figure 4a). This indicates that the microbial metabolism in soils beneath the decomposing wood was P-limited, and soil beneath the angiosperm wood had a higher degree of P limitation for microbial metabolism compared to the soil beneath gymnosperm wood in this study [42,44]. This finding could be explained by the positive correlation between the soil enzyme vector angle and wood decomposition rate (Figure S3). This is because fungi or other microorganisms could immobilize wood nutrients in the initial decay stage, and they also absorb nutrients from the underlying soils at the same time [22]. Thus, a higher wood decomposition rate results in larger microbial biomass and higher microbial activity in the decomposing wood, and this may put more pressure on the organic P mineralization of the underlying soils. Moreover, as angiosperm and gymnosperm wood differ greatly in their decomposition rates, we speculate that the degree of soil resource limitations beneath decomposing wood of other angiosperm and gymnosperm species may differ as well, which needs to be considered in future studies. However, the soil enzyme vector lengths did not differ significantly between the soils beneath the decomposing angiosperm and gymnosperm wood (Figure 4b), indicating that the soils had similar degrees of C limitation for microbial metabolism.

## 5. Conclusions

This study demonstrated that the tree types (using just four angiosperm and four gymnosperm species) of decomposing wood affected the enzyme activities but not the nutrient concentrations of underlying forest soils. In contrast to our first hypothesis, this indicates that differences in wood traits (i.e., in nutrient contents) do not affect the nutrient concentrations of underlying soils. In addition, this indicates that the tree types of decomposing wood have significant effects on underlying soil enzyme activities, but not through the correlations between soil nutrients and enzymes, as reported in previous studies. Moreover, the higher vector angle degree in the soils beneath angiosperm wood compared to gymnosperm wood indicates that soil microbial metabolism was more P-limited beneath decomposing angiosperm wood than gymnosperm wood. This may be caused by the higher decomposition rate of angiosperm wood, which leads to higher microbial activity and thus more pressure on organic P mineralization of underlying soils, compared to that of gymnosperm wood. Furthermore, as angiosperm and gymnosperm woods differ greatly in their decomposition rates, we speculate that the degree of soil

resource limitations beneath the decomposing wood of other angiosperm and gymnosperm species may differ as well, which needs to be considered in forest management and future studies. In summary, our findings highlight two points. First, the tree types of decomposing wood did not directly influence underlying soil nutrients but significantly affected soil enzymes. Second, soil enzyme activities beneath decomposing wood were not correlated with nutrient concentrations, but soil enzyme activity characteristics (e.g., its vector angle) may be related to the decomposition process of deadwood (e.g., its decomposition rate), as deadwood microorganisms could absorb nutrients from the underlying soils by improving nutrient release. These findings could improve our ability to accurately predict the role of wood decomposition on soil microbial ecology and nutrient cycles in forest ecosystems.

**Supplementary Materials:** The following supporting information can be downloaded at: https://www.mdpi.com/article/10.3390/f14091846/s1, Figure S1: Effects of decomposing wood on soil C/N ratio; Figure S2: Effects of decomposing wood on soil pH; Figure S3: Effects of tree types on wood decomposition rate (a) and wood decomposition rate in relation to soil enzyme vector angle (b); Table S1: Soil enzymes loadings on the first two axes of the principal component analysis.

**Author Contributions:** Z.-H.H., X.-Y.J. and Q.X. designed the experiments. X.-Y.J., Q.X. and Z.-Q.Z. performed the experiments. X.-Y.J. analyzed the data. X.-Y.J. wrote the paper. Z.-H.H., L.D. and Y.-X.Z. reviewed and edited the paper. All authors have read and agreed to the published version of the manuscript.

**Funding:** This research was supported by the National Natural Science Foundation of China (32271853), the GuangDong Basic and Applied Basic Research Foundation (2022A1515010663) and the Shaanxi Forestry Science and Technology Innovation Project (SXLK2022-05-3).

**Data Availability Statement:** The data presented in this study are available upon request from the corresponding author.

**Conflicts of Interest:** The authors declare no conflict of interest.

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
