# Peer review of "Influences of Wood Decomposition Associated with Tree Types on Soil Nutrient Concentrations and Enzyme Activities"

_forests, doi:10.3390/f14091846_

Round 1

Reviewer 1 Report

Review of “Influences of wood decomposition associated with tree species on soil nutrient concentrations and enzyme activities”

This paper describes a 3-year study to explore how different types of wood (four angiosperms and four gymnosperms) influence soil nutrients and enzyme activities beneath the wood. Throughout the authors use “tree species” to discuss this influence, but it might be better to define “tree type” early on and use that term instead throughout (even in the title: change to ““Influences of wood decomposition associated with tree types on soil nutrient concentrations and enzyme activities”). One limitation of this study is that only four species were used to represent all angiosperms and only four species were used to represent all gymnosperms, so the authors should take this limitation into account as they revise their discussion. The inferences they can make are limited by their study design.

I have many edits to provide, and will need to provide them as notations on a pdf because I do not have time to type out each change line by line and I was not provided with a word document to use for tracked changes. Please see the notations on the pdf. If the authors have any questions about my notations, please let me know.

The authors have a habit of mentioning the names of the authors of papers they cite – this makes reading their paper more challenging. In general, you should only mention the name of the authors if the people who wrote the paper are more important than the results of the study. Much of their introduction and discussion should be re-written with the authors names only in the citation and the work instead summarized in the sentence. Also, sometimes the authors use scientific names and sometimes they use common names – they should be consistent and use one or the other (or both, then just one or the other) throughout.

Line 69: which species were used in this study?

Line 84: shouldn’t this have said P-limited? Add citation #49?

Lines 85-92: please separate the enzyme vector model information into a new paragraph – provide much more context for this work – many readers will not have heard of these methods and they need more context and background information.

Lines 122-123: Add authorities after each scientific name (like provided in lines 126-128)

Line 164: spell out MUB

Figure 1: The authors should use a graphing method that doesn’t hide all data behind bars – violin plots are a good alternative. The results text lines 199-202 do not match the figure. Angiosperms do not differ from bare soil or from gymnosperms (according to the Tukey letters in panel a).

Lines 222-226: Add the acronyms for each enzyme as you introduce them here so you can use the shortened acronym in subsequent sentences

Figure 2: remove the asterisk in panel b if none of the treatments were significantly different.

Line 252: start a new paragraph about the vector angles and lengths – provide a lot more context about what these values mean.

Figure 4 legend: change to “Effects of decomposing wood on vector angle and vector length of soil enzymes based on the enzyme vector model.”

Figures 3 and 4: why are these not bar charts? Again, make violin plots for each one: Figure 3 panels b and c; Figure 4 panels a and b.

Lines 324-339: This section of the results is hard to read, again because of the focus on the authors of each paper instead of summarizing the patterns and explicitly comparing the results of this study to those prior studies.

Conclusions: The authors have written a conclusions section that is just a summary of their study, not a conclusion that explains why this is important. This needs a revision.

Check that all journal article titles have only the first word capitalized: problems on lines 389, 395, 432, 439, 465, 468, 471

Italicize Cunninghamia lanceolata on line 472

I have many edits to provide, and will need to provide them as notations on a pdf because I do not have time to type out each change line by line and I was not provided with a word document to use for tracked changes. Please see the notations on the pdf. If the authors have any questions about my notations, please let me know.

Reviewer 2 Report

The review of the paper entitled “Influences of wood decomposition associated with tree species on soil nutrient concentrations and enzyme activities” is complete and the report is given as below:

Research on the impact of wood decomposition by different tree species in regulating the soil nutrient concentration and enzyme activities is utmost important from both environmental and agricultural point of views. The paper is nicely written and the authors have added some valuable information in this domain. However, the paper can reach a greater audience if it is being modified in following ways:

1.     Information on C/N ratio could provide a robust information on soil microbial activity and thereby which could provide a clear understanding on the upregulation and down-regulation of soil enzymes and its concentration.

2.     LSD and p values are missing in the tables and figures which needs to be incorporated.

3.     Conclusion: What is the technical value of your knowledge gain? Please highlight it.

4.      A thorough check in the draft to publish the manuscript free of grammatical errors.

Grammatical and typographic errors needs to be checked thoroughly.

Reviewer 3 Report

General comments

This paper constitutes a solid contribution to the understanding of how decomposing wood affects soil nutrients and enzyme activities, an integral part of forest ecology. The authors deserve praise for their in-depth reach into this subject, substantiated by a well-executed literature review. The clear hypotheses presented at the beginning are admirable, effectively guiding the research and providing a robust framework for an objective evaluation of the results. The methods used in this study are pertinent and generally well-detailed, though it is suggested that the authors offer more explicit elaboration on the methodologies used. This would ensure that the study is accessible to readers who might not be familiar with these specific methods. Additionally, a more expanded discussion on potential sources of error or variation in the experiment would be valuable. For instance, addressing elements such as the sample size, the diversity of tree species, or any other environmental parameters that might not have been controlled in the experiment could fortify the research. The conclusion and discussion sections are well-articulated and competently place the findings within the context of existing research. However, a deeper exploration of the implications of the research is encouraged. This could involve potential practical applications or how this study could inform future research in the field. Lastly, some parts of the paper are somewhat dense with technical terminology. While this is understandable considering the nature of the research, a more accessible language would expand the appeal of the paper and render it more inclusive for a diverse audience.

Specific comments

Abstract

1.      The abstract, overall, is well-composed and comprehensive, but minor revisions could enhance clarity and impact. The language might be complex for a general audience. For instance, "C- (i.e., β-1,4-glucosidase, cellobiohydrolase), N- (i.e., N-acetyl-glucosaminidase), and P- (i.e., acid phosphatase) acquiring enzymes activities" might be simplified for better understanding, especially considering this is an abstract serving as an initial touchpoint for readers. The sentence "While how wood decomposition influences soil nutrient concentrations and enzyme activities associated with different tree species is not well understood" could be rephrased for improved readability.

Introduction

2.      The introduction, albeit comprehensive, might gain from a better thematic organization. A flow from the general role of deadwood to the differences between angiosperms and gymnosperms, then onto soil enzymes, and finally nutrient limitations in subtropical forests could improve the narrative and help readers better follow the argument's progression.

3.      Briefly stating the study's significance and novelty early on could assist the reader in understanding the unique contribution of the study in the existing body of knowledge.

4.      A detailed background on deadwood's role in nutrient cycling and carbon storage in forest ecosystems is given. However, the link between this background information and the specific study could be more explicitly drawn.

5.      Some concepts and processes, such as the enzyme vector model, are introduced without sufficient explanation. Additional sentences to simplify it for unfamiliar readers could be beneficial. Similarly, the concept of "vector angle degree" might need more clarification, especially for non-specialist readers.

6.      The hypotheses are clearly stated. However, the reasoning or literature support that led to these hypotheses' formulation could be strengthened and clarified. The hypotheses could be more specifically tied to the methods used, such as what kind of enzyme activities are focused on, or which nutrients are considered.

Materials and Methods

7.      The methods employed are apt for the research questions, but more detailed information about the experimental setup could be beneficial. More specifics about the tree species choice and the reasons behind selecting these particular angiosperm and gymnosperm species would add clarity. Were they representative of the dominant tree species, or was there another reason for their selection? Were they chosen due to their abundance, known differences in decomposition rates, or another factor? More details about the forest's history and management would be beneficial. How was the age of the trees determined? Why was C. fargesi chosen from non-plantation forest trees while all others were selected from plantation forest trees? This selection might introduce some variations in the results, which could be acknowledged in the limitations. Also, indicating how many wood samples were taken from each species and whether steps were taken to mitigate variations between different parts of the tree (e.g., sapwood vs. heartwood) would be helpful. More details on determining wood C, N, and P contents would also be valuable. More details on the sample size for soil samples would enhance the clarity and reproducibility of this study.

8.      More detail on how the enzyme activities were measured could be provided. The reasons behind selecting these specific enzymes could be better explained. Are they indicative of a particular process or function in the ecosystem?

9.      The statistical methods used seem appropriate, but more detailed information about the statistical tests and their assumptions would be beneficial. Any possible limitations due to these statistical methods should also be acknowledged. The authors could clarify why a PCA was specifically chosen for visualizing overall variation in soil enzymes.

Results

10.   Are there potential limitations or biases in the methodology or analysis that might have affected the results?

Discussion

11.   In the discussion, contrasting results found in previous studies are cited. How are the authors' findings reconciled with these contrasting studies?

12.   It is mentioned that there were no significant differences in soil total C and nutrient concentrations between angiosperm and gymnosperm wood. Was this consistent across all experimental trials, or were there instances where significant differences were observed?

13.   Discussing the reasons behind contrasting findings more deeply to identify gaps in the current understanding and suggest possible future research directions might be beneficial.

14.   The authors mention a significant difference in enzyme activities beneath angiosperm and gymnosperm wood. Could these differences be attributable to other environmental factors, such as differences in microbial community structure?

15.   Could the authors further explain their findings' implications on soil nutrient availability in the long term, considering the role of enzymes in nutrient cycling?

16.   Could the authors provide more context on the vector angle degree in the soils beneath angiosperm wood and its implications for microbial metabolism? Is this finding unique to the species used in the study, or can it be generalized across angiosperms?

Conclusions

17.   The authors frequently use "could partly explain," which somewhat weakens the authority of their statements. While scientific conclusions are rarely absolute, stronger language might more effectively communicate their interpretations.

18.   The conclusions are generally well-drawn, but they could further elaborate on the implications of these findings. Specifically, how does this new knowledge about the effects of tree species on soil enzyme activities enhance understanding of nutrient cycles in forest ecosystems? How can it help improve forest management practices, especially in managing wood decomposition?

19.   The authors briefly note that their work can help improve predictions about the role of wood decomposition on soil microbial ecology and nutrient cycles, but they could expand on how these predictive capabilities might be enhanced or the specific applications they might have.

20.   How do the findings contribute to the broader understanding of carbon and nutrient cycling in forest ecosystems?

21.   Are there plans to extend this study over a longer time frame? If so, what changes or additions to the methodology would be anticipated?

22.   The authors could suggest future experiments or data collection that might be required to further validate their conclusions. Including potential challenges or limitations in the field that could impact the validity of their findings or impede their application would also be a worthy addition to this conclusion. This would provide a more comprehensive perspective of the research's overall impact.

The quality of English in the research paper appears to be mostly good, with clear and coherent sentences. The authors provide an adequate explanation of the research and its context. However, a minor level of editing could still improve the readability and flow of the text. Overall, the authors have done a commendable job in presenting their research in a professional, scholarly manner.

Round 2

Reviewer 1 Report

Review of revision – “Influences of wood decomposition associated with tree types on soil nutrient concentrations and enzyme activities”

The authors did a nice job of revising the paper and it is now in an improved state. A few new grammatical errors were introduced in new revised text and the authors should consider these edits carefully.

One aspect of the revision that needs a bit more work is to better qualify the results based on the limited number of species involved in the study to address the question of comparing gymnosperms and angiosperms.

Line numbers match the "tracked changes" version of the revision

Line 13: change to “Wood traits that vary by tree species can influence decomposition…”

Line 16: change to “…how tree type (for four angiosperm vs. four gymnosperm species) affects underlying soil total carbon…”

Line 21: change to “…decomposing wood were not different between angiosperm and gymnosperm species”

Line 22: change to “…P-acquiring  enzymes…”

Line 23: change to “between angiosperm and gymnosperm species”

Line 28: change to “…decomposing wood may affect underlying soil enzyme activities…”

Line 217: change to “…which may acidify soils [18].”

Line 219: change to “…compared to broadleaf forests.”

Line 229: change to “…soil nutrient concentrations or microbial biomass”

Line 235: change to “Moreover, resource allocation theory posits…”

Line 421: change to “…gymnosperm wood [10], and more released dissolved organic C…”

Line 425: change to “…(i.e., hypothesis 1), and the regulatory effect of soil nutrient…”

Line 576: change to “…further calculated using the vector length and vector angle…”

Line 582: change to “The fluorescent substrates were…”

Line 615: change to “T-tests were used to compare…”

Line 641: change to “…(both p > 0.05; Figure 1b, 1e), or soil C:N ratio…”

Line 858: change to “Three factors could explain…”

Line 863: Add a sentence like this: “Third, the eight species chosen do not show differences between gymnosperms and angiosperms and the extent of the study should be expanded.”

Then line 863: change to “Our findings suggest that tree types for these eight species of decomposing wood…”

Line 867: change to “A previous study based on 30 forest plots…”

Line 986: change to “…the stage of decomposition in some studies was much more important than tree species…”

Line 991: change to “…tree species and tree types may have a limited effect on soil nutrient…”

Line 996: change to “…angiosperm and gymnosperm species. Soils beneath angiosperms…”

Line 998: change to “…while soils beneath gymnosperm wood…”

Line 1002: change to “…P addition treatment can reduce soil N…”

Line 1003: change to “…this mechanism does not fully explain the differences in soil…”

Line 1004: change to “…total C and nutrient concentrations did not differ beneath decomposing wood between angiosperm and gymnosperm species”

Line 1011: change to “…resources in soils beneath…”

Line 1015: change to “…angiosperm and gymnosperm species…”

Line 1018: change to “A previous study showed that…”

Line 1031: change to “…NAG activities are usually negatively correlated…”

Line 1299: change to “…aligning with one previous study…”

Line 1301: change to “…angiosperm and gymnosperm species.”

Line 1302: change to “…activity patterns were affected by tree types…”

Line 1304: change to “…angiosperms and gymnosperms”

Line 1317: change to “…may differ as well, which needs to be considered…”

Line 1318: change to “However, the soil enzyme vector lengths…”

Line 1322: change to “…tree types (using just four angiosperm and four gymnosperm species)…”

See edits above
